# Targeting Energy Protection as a Novel Strategy to Disclose Di’ao Xinxuekang against the Cardiotoxicity Caused by Doxorubicin

**DOI:** 10.3390/ijms24020897

**Published:** 2023-01-04

**Authors:** Tao Wang, Chuqiao Yuan, Jia Liu, Liangyan Deng, Wei Li, Junling He, Honglin Liu, Liping Qu, Jianming Wu, Wenjun Zou

**Affiliations:** 1State Key Laboratory of Southwestern Chinese Medicine Resources, Chengdu University of Traditional Chinese Medicine, Chengdu 611137, China; 2School of Pharmacy, Chengdu University of Traditional Chinese Medicine, Chengdu 611137, China; 3School of Basic Medical Sciences, Southwest Medical University, Luzhou 646000, China

**Keywords:** cardiotoxicity, energy metabolism, mitochondrion, doxorubicin, Xinxuekang

## Abstract

Doxorubicin (DOX) can induce myocardial energy metabolism disorder and further worsen heart failure. “Energy protection” is proposed as a new cardiac protection strategy. Previous studies have found that Di’ao Xinxuekang (DXXK) can improve doxorubicin-induced cardiotoxicity in mice by inhibiting ferroptosis. However, there are very few studies associating DXXK and energy protection. This study aims to explore the “energy protection” effect of DXXK on cardiotoxicity induced by DOX. A DOX-induced cardiotoxicity model established in rats and H9c2 cells are used to analyze the therapeutic effects of DXXK on serum indexes, cardiac function indexes and cardiac histopathology. The metabonomic methods were used to explore the potential mechanism of DXXK in treating DOX-induced cardiotoxicity. In addition, we also observed the mitochondrial- and autophagy-related indicators of myocardial cells and the mRNA expression level of the core target regulating energy-metabolism-related pathways. Our results indicated that DXXK can improve cardiac function, reduce myocardial enzymes and alleviate the histological damage of heart tissue caused by DOX. In addition, DXXK can improve mitochondrial damage induced by DOX and inhibit excessive autophagy. Metabonomics analysis showed that DOX can significantly affects the pathways related to energy metabolism of myocardial cells, which are involved in the therapeutic mechanism of DXXK. In conclusion, DXXK can treat DOX-induced cardiotoxicity through the AMPK-mediated energy protection pathway.

## 1. Introduction

With the development of diagnostic and therapeutic techniques, the cardiotoxicity accompanied by the improved survival rate and survival time in patients with malignant tumors cannot be ignored. Epidemiological data shows that the risk of cardiovascular death of tumor patients increases by 1.6–3.6 times 5 years after diagnosis. Cardiovascular death caused by antitumor drugs became the second major cause of death after tumor recurrence and metastasis [1]. The risk of death associated with cancer has exceeded the risk of cancer recurrence, especially for patients with malignant tumors complicated with cardiovascular disease. Anthracyclines are widely used in the treatment of various malignant tumors and malignancies of the blood and lymphatic system, and up to 60% of cancer patients are treated with anthracyclines [2,3]. As the representative of the anthracyclines, the incidence of cardiac dysfunction caused by the representative drug DOX in the treatment of adult and child patients is about 30% and 60%, respectively. Patients tend to experience dose-dependent cardiotoxicity with prolonged chemotherapy cycles. Previous studies have reported that LVEF in patients is generally reduced when the DOX cumulative dose exceeds 350 mg/m^2^, heart failure occurs when the DOX cumulative dose exceeds 550 mg/m^2^ and HF prevalence is as high as 18–48% at 700 mg/m^2^ [4,5]. With the continuous accumulation of DOX, the proportion of patients with congestive heart failure gradually increases, showing irreversible cardiotoxicity [6,7,8]. Dexamethylamine is currently the only drug approved for the treatment of DOX-induced cardiotoxicity; it is expensive and increases the risk of secondary cancer in patients [9]. Similarly, due to the unknown preventive and therapeutic effects of β-blockers, coenzyme Q10 and statins, their clinical application is severely limited [10]. Therefore, cardiotoxicity caused by anthracyclines, especially DOX, has become an urgent clinical issue.

Having a strong affinity with cardiomyophospholipids, DOX can closely combine and accumulate with the myocardial cell membrane after passively passing through the sarcoplasmic membrane, accumulating in the myocardial cytoplasm, causing oxidative stress, inflammation, energy metabolism disorder, etc., and leading to permanent myocardial cell injury and cell death [11]. In addition, as the signaling pathways of DOX-triggered cardiotoxicity are interrelated, its core mechanism is still unclear. As there are both hydrophobic and hydrophilic domains in the molecular structure, DOX is characteristic of a phospholipid-loving molecule, which is, therefore, the most susceptible target organ. Studies have pointed out that DOX can not only disturb the normal transmission of the respiratory chain and the structure and function of creatine kinase in mitochondria but also that the concentration of DOX in mitochondria is 100 times that in plasma; induced bioenergy disorder is one of the markers of cardiotoxicity caused by DOX [12,13]. As we all know, normal energy supply is a necessary condition to ensure the normal physiological and biochemical process of myocardial cells and the material basis to maintain the stability of the internal environment of the heart and the diastolic and contractile functions of the heart [14,15]. The heart, the largest energy-consuming organ in the body, needs to synthesize adenosine-triphosphate (ATP)—an unstable high-energy compound that is the most direct energy source in organisms—continuously to meet needs [16]. DOX is closely combined with the myocardial cell membrane to continuously produce ROS through various ways such as electron transfer that accumulates in myocardial cells [17], which causes activation of oxidative stress, inflammation, etc., downregulating the activities of Ca^2+^ and ATPase inside and outside myocardial cells, limiting Ca^2+^ activity, and energy utilization efficiency, damaging mitochondria, leading to permanent myocardial cell injury and cell death. However, myocardial injury caused by DOX involves many biological processes such as inflammation, oxidative stress, lipid peroxidation and DNA damage [18], all of which can lead to myocardial energy metabolism disorder and form a vicious circle. About 60–90% of normal heart energy supply comes from fatty acid oxidation, about 10–40% from glycolysis, and the rest from ketone bodies and amino acids [19,20,21]. DOX-induced heart injury can greatly reduce the energy supply of fatty acid sources that glycolysis uses as the main myocardial energy supply; the decrease in the expression of key rate-limiting enzymes in glycolysis with the aggravation of injury may make metabolites accumulate in myocardial tissue, resulting in a reduced pH value, which will further downregulate the activity of glycolysis rate-limiting enzymes and then worsen the myocardial energy supply, eventually leading to heart failure [22,23]. In addition, studies have suggested that giving various cell death inhibitors alone has limited improvement on the survival rate of myocardial cells [24]. As energy is the most direct guarantee for the normal metabolic process of myocardial cells in various organisms, the strategy of “energy protection” has gradually gained researchers’ attention; it plays a protective role from the basic physiological and biochemical processes of the heart by regulating myocardial energy metabolism. 

DXXK is contained in the Pharmacopoeia of the People’s Republic of China 2020. It is a traditional Chinese medicine composed of Dioscorea nipponica Makino extract, mainly composed of pseudoprotodioscin, protodioscin and other ingredients. DXXK, which has been widely used to prevent coronary heart disease and angina pectoris for more than 30 years, has pharmacological effects such as expanding the coronary arteries and improving myocardial ischemia [25,26]. In recent years, clinical and pharmacological studies have shown that DXXK has good efficacy and a role in cardiovascular diseases, such as coronary atherosclerosis, myocardial ischemia reperfusion injury and myocardial infarction [27,28]. Studies have shown that DXXK can correct the abnormal mitochondrial membrane potential, clear the peroxides of myocardial cells and myocardial tissues, and protect mitochondria [29]. In addition, our previous research has confirmed that DXXK can protect the ferroptosis in mouse hearts caused by doxorubicin [30]; however, whether DXXK can improve doxorubicin-induced cardiac toxicity of rats through “energy protection” is still unclear. Metabonomics is an important part of systems biology as a research method to analyze all the metabolites in organisms and find the relationship between the metabolites and diseases. It can macroscopically control the disease change process from the whole biological signal network and then capture different metabolites between different groups. It can comprehensively reflect the biological mechanism of DOX-induced cardiotoxicity and the protective pathway of DXXK in myocardial cells and focus on the energy metabolism of myocardial cells.

## 2. Results

### 2.1. DXXK Protects against Heart Injury Caused by DOX

Our results show that when the DOX dose accumulates to 15 mg/kg, it can cause significant heart damage, which is mainly manifested by the increase in serum myocardial enzyme content, a large amount of inflammatory cell infiltration in heart tissue and vacuoles and watery lesions in cardiac cells (Figure 1A–E). Administration of DXXK significantly reduced the serum myocardial enzyme level of rats, improved the histopathological changes of the rat heart, eliminated the vacuole and watery lesions in myocardial cells and only a small amount of inflammatory cell infiltration was observed, indicating that DXXK protected the heart from DOX-induced cardiotoxicity (Figure 1A–E).

### 2.2. DXXK Improves DOX-Induced Cardiac Function Decline and Electrical Signal Conduction Block in Rats

We observed the echocardiography of rats and found that when the cumulative dose of DOX reached 15 mg/kg, compared with the control group, the DOX group had significantly reduced heart rate, LVEF, LVFS, SV and CO, and DXXK had a significant improvement effect on the above indicators (Figure 2A–E). In addition, the ECG of rats in the DOX group also showed obvious abnormalities. DOX can cause the prolongation of the PR interval, QT interval, QTC interval, QRS duration and the increase in the ST time period, which indicates that DOX causes cardiac electrical signal conduction block in rats, whereas the increase in the ST time period also indicates that myocardial ischemia may exist in rats (Figure 2F,G). Similarly, DXXK has also played a significant role in restoring cardiac function and correcting abnormal cardiac electrical signal conduction.

### 2.3. DXXK Protects H9c2 Cells from DOX-Induced Injury

Our results show that 4 μmol DOX can significantly inhibit the survival rate of H9c2 cells (rat heart myoblast cell line) and significantly increase the level of oxidative stress in H9c2 cells. DXXK with doses of 800 ng/mL, 400 ng/mL and 200 mg/mL significantly improved the survival rate of H9c2 cells (Figure 3A) and reduced the oxidative stress level in H9c2 cells (Figure 3B–D). In addition, we used flow cytometry to detect the ROS level in H9c2 cells and found that DOX significantly increased the ROS level in H9c2 cells, whereas DXXK increased the ROS level in H9c2 cells in a dose-dependent manner (Figure 3E). As the main production site of ROS, our results not only indicate that DOX may induce myocardial mitochondrial damage but also suggest that DXXK may have a protective effect on myocardial mitochondria.

### 2.4. DXXK Protects Myocardial Mitochondria from DOX-Induced Damage

ATP is the most direct source of energy for cardiac myocytes. We first measured the ATP content of cardiac myoblasts. The results show that, as expected, the ATP content in H9c2 cells was significantly reduced under DOX intervention, which induced the energy failure of cardiac myocytes. DXXK at three doses significantly improved the above situation. As we all know, mitochondria are the main site of ATP production. Therefore, in order to explore the cause of ATP depletion, we observed the number of mitochondria and autophagosomes in myocardial cells through laser confocal observation (Figure 4A,B). We found that DOX can significantly reduce the number of mitochondria in myocardial cells and significantly increase the content of autophagosomes in myocardial cells. This suggests that the depletion of ATP in myocardial cells may be related to the decrease in the number of mitochondria. In order to make up for energy depletion, autophagy of cardiomyocytes is activated competitively. DXXK significantly restored the number of mitochondria in cardiomyocytes and inhibited excessive autophagy in cardiomyocytes (Figure 4C). In addition, we used a JC-1 kit to detect the mitochondrial membrane potential of myocardial cells. The results show that DOX can significantly reduce the membrane potential of myocardial mitochondria, which indicates that DOX has the effect of damaging mitochondrial function, whereas DXXK significantly increases the membrane potential of mitochondria (Figure 4D,E).

### 2.5. DOX Induces Metabolic Disorder in H9c2 Cells

In the positive and negative ion modes, we found from OPLS−DA and PLS−DA that the sample aggregation of the control group and DOX group was high and that the two groups were separated from each other, indicating that there was a metabolic difference between the control group and DOX group. Similarly, there were metabolic differences between the DOX and DXXK groups. In the positive and negative ion mode, there is no overlap and crossover between the samples from the control group, DOX group and DXXK group (Figure 5A–D). In addition, in order to find potential biomarkers, the relative standard deviation (RSD) of the characteristic peak in the sample usually does not exceed 30%. In this study, the proportion of characteristic peaks with RSD < 30% of samples in ESI+ and ESI− modes exceeded 80% of the total samples, indicating that the quality of samples in this study is high, the repeatability is good and it can better reflect the metabolic differences between groups (Figure 5E,F). In order to characterize the metabolism more comprehensively, we obtained the differential metabolites between the control group, DOX group and DXXK group under ESI+ and ESI− modes. Then, we further plotted the S-plots and determined the variables that play an important role in the cluster. Finally, we found that there were significant changes in the metabolism of myocardial cells after DOX and DXXK intervention.

### 2.6. Identification of Potential Biomarkers of DOX-Induced Cardiotoxicity and DXXK Cardioprotection

We identified the different metabolites between different groups and found that there were 18 different metabolites in the control group and DOX group, of which 9 metabolites were upregulated and 16 metabolites were downregulated. There were 11 metabolites between the DOX group and the DXXK group, of which seven were upregulated and four were downregulated (Figure 6A,B). In order to further screen specific differential metabolites, we used VIP ≥ 1.5 and *p*-value ≤ 0.01 as screening conditions and found that there were 15 differential metabolites between the DOX and control groups (Table 1), whereas there were seven differential metabolites between the DOX and DXXK groups (Table 2). In order to characterize the potential relationship of these metabolites, we first observed the situation of the above differential metabolites in each sample (Figure 6C,D) and then carried out association analysis for these differential metabolites. The red represents that the two metabolite levels are positively correlated; the blue represents that the two metabolite levels are negatively correlated (Figure 6E,F).

### 2.7. Pathway Analysis of DXXK Protecting Myocardial Cells from DOX-Induced Injury

In order to further reveal the mechanism by which DXXK protects against DOX induced myocardial cell damage, we conducted enrichment analysis of KEGG pathways for the above differential metabolites. We found that there were 65 differential pathways between the DOX group and the control group and 29 differential pathways between the DOX group and the DXXK group. We demonstrated the top 20 signaling pathways (Figure 7A,B) and built a “signaling pathway-metabolite” network (Figure 8C,D). We found that these differential signaling pathways mainly focused on regulating amino acid metabolism, fatty acid metabolism, carbohydrate metabolism, phospholipid metabolism, glutathione metabolism, pantothenic acid and coenzyme A biosynthesis.

### 2.8. DXXK Promotes the Expression of Key Target mRNAs in the Myocardial Energy Metabolism Pathway

In order to further verify whether DXXK can protect H9c2 cells from damage through the energy pathway, we detected mRNA expression that can regulate targets related to energy metabolism. We found that DOX can significantly inhibit the mRNA expression of AMPK, PPAR-α, CPT-1, GLUT4 and CD36 and significantly increase the mRNA expression of ACC; however, DXXK could correct the above process (Figure 8A–F). These results indicate that DOX can damage the utilization of fatty acid and glucose in myocardial cells, affect mitochondrial function and then induce energy depletion of myocardial cells. DXXK can improve the energy depletion status of myocardial cells by improving the above processes.

## 3. Discussion

The cardiotoxicity of anthracyclines is characterized by dose dependence, progression and irreversibility. The incidence of acute cardiotoxicity caused by anthracyclines was 1.0%, chronic cardiotoxicity 1.6~2.1% and delayed cardiotoxicity 1.6~5.0% [31]; especially for DOX, the most typical drug [32]. The complex pathological mechanism and influencing factors for DOX-induced cardiotoxicity are involved with hemodynamics, neurohormones, energy and lipid metabolism, etc. However, the current research mainly focuses on cardiac inflammation, oxidative stress, apoptosis and iron death caused by DOX [33]. Professor Neubauer believes that insufficient myocardial energy supply or metabolic imbalance can damage heart structure and function, thus further inducing heart failure [34]. The heart, also called the “fuel engine”, of which disordered energy metabolism can cause abnormal structure and function changes, according to the concept of “metabolic remodeling” put forward by Professor Van Bilsen. Obviously, energy protection and improvement of cardiac metabolic disorder is one of the new effective ways to treat DOX-induced cardiotoxicity [35]. 

As a serine/threonine protein kinase, AMP-activated protein kinase (AMPK), which widely exists in myocardial cells, is an energy receptor of cells and participates in the regulation of cell energy metabolism to play an important role in physiological and pathological conditions [36,37]. The activated AMPK phosphorylation regulatory protein can not only increase ATP production and the catabolic pathway and the inhibit energy storage pathway (synthesis of lipids, fatty acids, cholesterol and proteins) during myocardial ischemia, hypoxia and strenuous exercise but can also directly regulate the transcription of metabolic enzymes and genes to control energy metabolism [38]. When the ratio of AMP/ATP increases, AMP binds to the γ subunit with regulatory activity to activate AMPK when the decreased production and increased consumption of ATP is caused by ischemia, hypoxia, stress and muscle contraction [39,40]. Activated AMPK can phosphorylate a variety of downstream target proteins that are involved in myocardial metabolism, fibrosis and oxidative stress [41]. Therefore, energy metabolism regulated by AMPK can be used as one of the prevention and treatment targets for DOX-induced cardiotoxicity. 

Myocardial metabolism varies significantly during heart injury, including changes in substrate utilization and mitochondrial dysfunction. Usually, glucose is the main substrate used by myocardial mitochondria because glucose produces more energy under the same oxygen consumption [42]. Recent studies have shown that with the aggravation of heart injury, the utilization rate of ketones and branched chain amino acids in myocardial cells increases, as well as there being increased ATP production by increasing the level of acetyl-coenzyme A [43]. Activated AMPK can participate in the regulation of myocardial metabolism and mitochondriogenesis. As a medium for the translocation, AMPK can improve the transport efficiency of glucose transporter 4 (GLUT4, a key protein in glucose metabolism) from the cytoplasm to the cell membrane and keep GLUT4 at its active site, to increase glucose metabolism levels in the myocardium [44]. AMPK can also increase the intake of fatty acids per unit time of myocardial cells and the activity of acetyl-coenzyme A carboxylase by improving the translocation speed of fatty acid transporter CD36 and increase to decrease the concentration of malonyl-coenzyme A, thus improving the activity of carnitine palmitoyltransferase I and the oxidation of fatty acids [45]. In addition, AMPK can activate peroxisome proliferator-activated receptor γ coactivator-1α (PGC-1α) by directly phosphorylating serine or increase the activity of PGC-1α by indirect deacetylation, inducing mitochondrial biogenesis [46]. Therefore, AMPK can improve the myocardial metabolism of DOX-induced cardiotoxicity by regulating glucose metabolism, fatty acid metabolism and mitochondrial biogenesis during myocardial injury. 

Moreover, as a way to maintain cell homeostasis and save energy, moderate autophagy can restore the structure and function of the heart during heart injury, whereas AMPK inhibits cardiac hypertrophy and delays the progress of cardiac injury by promoting autophagy [47]; AMPK and autophagy are related to the functional state of the heart. Activated AMPK can reduce autophagy in a failed heart but induce autophagy in non-failed hearts. Activated AMPK can regulate autophagy by enhancing Unc-51 like kinase 1 (ULK1, a key enzyme) to regulate autophagy activity through two different mechanisms [48]. Firstly, AMPK can increase the phosphorylation of ULK1 by reducing the activity of mammalian rapamycin target protein [49]. Secondly, it can enhance the activity of ULK1 by direct phosphorylation of four residues, SER467, SER555, THR574 and SER637, to regulate autophagy [50]. 

Therefore, we used AMPK as the core to detect multiple targets that can regulate myocardial energy metabolism and found that DXXK can significantly upregulate the mRNA level of AMPK, GUT4, CD36, PPAR and CPT-1 in myocardial cells, indicating that DXXK can protect against cardiotoxicity by regulating the process of myocardial energy metabolism. Interestingly, DXXK did not alleviate energy depletion of myocardial cells by activating AMPK to promote autophagy in myocardial cells but inhibited excessive autophagy of myocardial cells. Therefore, DOX can inhibit the energy generating function of AMPK while inducing the energy deficiency of cardiomyocytes, leading to the depletion of myocardial energy. DXXK can relieve DOX’s inhibition of AMPK and promote the energy generation of cardiomyocytes. DXXK not only upregulates CD36 to transport fatty acids that can be directly utilized by mitochondria and synthesize ATP to cardiomyocytes but also activates CPT-1 to promote the utilization of fatty acids by mitochondria. DXXK also promotes glucose to enter cardiomyocytes by upregulating GLUT4, thereby alleviating the energy generating substrate deficiency caused by DOX. In addition, DXXK also plays a protective role in mitochondria by upregulating PPAR and promotes mitochondrial production, indicating that DXXK can interfere with multiple links of energy metabolism and fight against DOX-induced cardiotoxicity (Figure 9). However, previous studies have reported that AMPK can inhibit mTOR, promote ULK1 expression and activate the autophagy of cardiomyocytes [51]. Therefore, we speculate that DXXK participates in the process of regulating the autophagy of cardiomyocytes through a non-AMPK pathway. In addition, although this study confirmed that DXXK can increase the ATP content of myocardial cells by promoting the uptake and utilization of energy substrates by mitochondria, it is still unclear how DXXK regulates the ATP production process in mitochondria. We will carry out further research in this area in the future. Previous studies have found that Di’ao Xinxuekang (DXXK) can improve doxorubicin-induced cardiotoxicity in mice by inhibiting ferroptosis. In this study, a model of doxorubicin-induced cardiotoxicity was established in rats and a new interpretation of DXXK’s cardioprotective effect was suggested from the perspective of energy metabolism.

## 4. Materials and Methods

### 4.1. Animal Handling

Male SD rats weighing 200 ± 20 g (License No. SCXK 2019-0008) were provided by Beijing Scientific Animal Breeding Center (Beijing, China); the quality control department is the National Institute of Food and Drug Administration. All operations on rats were performed in accordance with the approved guidelines of the Animal Experimentation Ethics Committee at Chengdu University of Traditional Chinese Medicine (License NO. TCM-2016-312). The cardiotoxicity model was established by doxorubicin hydrochloride injection (2.5 mg/kg, cumulative 15 mg/kg) and the rat ECG signal was evaluated using a RM6240 series multi-channel physiological signal acquisition instrument. The rats were randomly divided into three groups: (A) Control group; (B) DOX group; (C) DOX + DXXK group. Rats in the DXXK group were given DXXK 200 mg/kg by gavage once a day for 42 consecutive days. After the last gavage, the rat heart function was tested and the rat serum samples and heart tissues were collected and stored at −80 °C for subsequent testing.

### 4.2. Detection of Pharmacodynamic Indexes

Serum levels of CK-MB, LDH, AST and ALT were detected using a BS-360S automatic serum biochemical analyzer (Mindray, Shenzhen, China). The left ventricular myocardium was cut longitudinally and fixed with 4% paraformaldehyde. Myocardial morphological changes were detected by heme-eosin (H&E). The paraffin-embedded sections were observed under a Nikon microscope and PRO PLUS 7200 software. The ECG signals of animals were collected and analyzed by the BL-420N biological signal acquisition and analysis system. 

### 4.3. Reagents and Antibodies

H9c2 cardiomyoblasts were cultured in DMEM containing 10% (*v*/*v*) FBS and 1% penicillin–streptomycin at 37 °C in a moist incubator with 5% CO_2_. The cells were subcultured when cell density was about 80–90% and divided into control group, DOX group, DXXK 800 ng/mL group, DXXK 400 ng/mL group and DXXK 200 ng/mL group. The DOX group was given DOX 4 μ mol/mL for 24H; the DXXK groups were given DOX 4 μ mol/mL + DXXK 800 ng/mL, DOX 4 μ mol/mL + DXXK 400 ng/mL and DOX 4 μ mol/mL + DXXK 200 ng/mL, respectively. H9c2 rat cardiomyocytes were purchased from GIBCO; THERMO FISHER SCIENTIFIC, INC.; FBS (Beyotime, Shanghai, China), penicillin–streptomycin (Beyotime, Shanghai, China) and DMEM (Beyotime, Shanghai, China). 

### 4.4. Cell Survival Rate

The cells were counted after digesting cardiomyocytes with trypsin and prepared into a certain concentration of cell suspension, which was inoculated into a 96-well plate at 8 × 10^3^ cells/well, 100 μL per well. After adhering to the wall, the cells were intervened with drugs, with each plate provided with a blank well, a control well and experimental wells. The experimental wells contained cells, complete culture medium, drug solution with corresponding concentration and CCK-8 solution (Beyotime, Shanghai, China). The control well contained cells, complete culture medium and CCK-8 solution, without drug solution. The blank well contained complete culture medium and CCK-8 solution, without cells and drug solution. After drug intervention, 10 μL of CCK-8 solution was added to each well, the 96-well plates were incubated in an incubator for 30 min, the absorbance at 450 nm was measured by microplate reader and the cell survival rate was calculated. 

### 4.5. Oxidative Stress Detection

H9c2 cardiomyocytes were inoculated into a 12-well plate with 5 × 10^5^ cells/well for 24 h and treated with DXXK solution and DOX (4 μM) for 24 H. There was a control group, DOX group, DXXK 800 ng/mL group, DXXK 400 ng/mL group and DXXK 200 ng/mL group, each with 6 double wells. The medium-effective RIPA cell lysate after incubation was mixed with protease inhibitor and phosphatase inhibitor in a ratio of 100:1 to obtain cell lysate. Proteins were quantified according to BCA kit (Bicinchoninic Acid Assay), such as the contents of SOD, CAT and MDA of myocardial cells in each group. DCFH-DA was diluted with DMEM at 1:1000 to a probe solution with a final concentration of 10 μmol/L. A total of 1 mL DCFH-DA solution was added to each well after removing the culture medium and the plates incubated at 37 °C for 20 min before washing with DMEM 3 times to remove DCFH-DA that did not enter the cells. Then, the solution was collected following trypsin digestion and blown into a single-cell suspension by adding PBS. Flow cytometry (488 nm excitation wavelength, 525 nm emission wavelength) was used to detect the ROS level of each group of cells according to fluorescence intensity. 

### 4.6. Detection of Mitochondria and Lysosomes by Laser Confocal Method 

JC-1 stain was prepared according to the JC-1 instructions; 100 μL of which was added to each dish after rinsing with PBS 2 times, mixing evenly, and incubating at 37 °C for 20 min. Then, the supernatant was removed and washed twice with JC-1 staining buffer (1×). After adding 2 mL culture medium to each dish, the maximum excitation and emission wavelengths of the JC-1 monomer were 514 nM and 529 nM, respectively. The maximum excitation and emission wavelengths of the JC-1 polymer are 585 nm and 590 nm, respectively. Therefore, the mitochondrial and lysosomal contents were detected by laser confocal technology, according to the above operation steps and the requirements of LYSOTRACKER RED of the Solarbio Company (Beijing, China) and MITO-TRACKER GREEN of the Solarbio Company (Beijing, China). ATP content was detected in terms of the requirements of the Solarbio kit.

### 4.7. Extraction of Intracellular Metabolites

Transfer all cell samples into a 2 mL centrifuge tube; add 100 mg glass beads. Accurately add 1000 µL acetonitrile (ACN): methanol: H_2_O mixed solution (2:2:1, *v*/*v*/*v*) (stored at 4 °C), vortex for 30 s. Put the centrifuge tube containing the sample into the 2 mL adapter matched with the instrument, immerse it in liquid nitrogen for rapid freezing for 5 min, take out the centrifuge tube and thaw at room temperature, put the centrifuge tube into the 2 mL adapter again, install it into the tissue grinder and grind it at 55 Hz for 2 min. Repeat step 3 twice. Take out the centrifuge tube, centrifuge for 10 min at 12,000 rpm and 4 °C, take all the supernatant, transfer it to a new 2 mL centrifuge tube and concentrate and dry it. Accurately add 300 µL acetonitrile: 2-amino-3-(2-chloro-phenyl)-propionic acid (4 ppm) solution prepared with 0.1% formic acid (1:9, *v*/*v*) (stored at 4 °C) to redissolve the sample, filter the supernatant by 0.22 µm membrane and transfer into the detection bottle for LC-MS detection.

### 4.8. RUPLC-Q-TOF-MS Conditions

The LC analysis was performed on an ACQUITY UPLC System (Waters, Milford, MA, USA). Chromatography was carried out with an ACQUITY UPLC ^®^ HSS T3 (150 × 2.1 mm, 1.8 µm) (Waters, Milford, MA, USA). The column was maintained at 40 °C. The flow rate and injection volume were set at 0.25 mL/min and 2 μL, respectively. For LC-ESI (+)-MS analysis, the mobile phases consisted of (C) 0.1% formic acid in acetonitrile (*v*/*v*) and (D) 0.1% formic acid in water (*v*/*v*). Separation was conducted under the following gradient: 0~1 min, 2% C; 1~9 min, 2~50% C; 9~12 min, 50~98% C; 12~13.5 min, 98% C; 13.5~14 min, 98~2% C; 14~20 min, 2% C. For LC-ESI (-)-MS analysis, the analytes were carried out with (A) acetonitrile and (B) ammonium formate (5 mm). Separation was conducted under the following gradient: 0~1 min, 2% A; 1~9 min, 2~50% A; 9~12 min, 50~98% A; 12~13.5 min, 98% A; 13.5~14 min, 98~2% A; 14~17 min, 2% A. Mass spectrometric detection of metabolites was performed on a Q Exactive (Thermo Fisher Scientific, Waltham, MA, USA) with an ESI ion source. Simultaneous MS1 and MS/MS (Full MS-ddMS2 mode, data-dependent MS/MS) acquisition was used. The parameters were as follows: sheath gas pressure, 30 arb; aux gas flow, 10 arb; spray voltage, 3.50 kV and −2.50 kV for ESI (+) and ESI (−), respectively; capillary temperature, 325 °C; MS1 range, *m/z* 81–1000; MS1 resolving power, 70,000 FWHM; number of data dependent scans per cycle, 10; MS/MS resolving power, 17,500 FWHM; normalized collision energy, 30%; dynamic exclusion time, automatic.

### 4.9. Data Statistics and Analysis

The raw data were firstly converted to mzXML format by MSConvert in the ProteoWizard software package (v3.0.8789) and processed using XCMS for feature detection, retention time correction and alignment. The metabolites were identified by accuracy mass (< 30 ppm) and MS/MS data which were matched with HMDB (http://www.hmdb.ca, accessed on 14 September 2022), massbank (http://www.massbank.jp/, accessed on 14 September 2022), LipidMaps (http://www.lipidmaps.org, accessed on 15 September 2022), mzcloud (https://www.mzcloud.org, accessed on 16 September 2022) and KEGG (http://www.genome.jp/kegg/, accessed on 16 September 2022). The robust LOESS signal correction (QC-RLSC) was applied for data normalization to correct for any systematic bias. After normalization, only ion peaks with relative standard deviations (RSDs) less than 30% in QC were kept to ensure proper metabolite identification. Ropls software was used for all multivariate data analyses and modelings. After scaling data, models were built on orthogonal partial least square discriminant analysis (PLS-DA) and partial least-square discriminant analysis (OPLS-DA). All the models evaluated were tested for overfitting with methods of permutation tests. Finally, *p* value < 0.05 and VIP values > 1 were considered to be statistically significant metabolites. Differential metabolites were subjected to pathway analysis by MetaboAnalyst, which combines results from powerful pathway enrichment analysis with the pathway topology analysis. The identified metabolites in metabolomics were then mapped to the KEGG pathway for biological interpretation of higher-level systemic functions.

### 4.10. PCR Detection

H9c2 myocardial cells, 5 × 10^5^ cells/well, were inoculated into 24-well plates, cultured for 24 h and treated with DXXK solution and DOX (4 µM) for 24 h. There were three groups: control group, DOX group and DXXK (400 ng/mL) group. Each group has 6 repeats. All cells were cultured in a 37 °C CO_2_ incubator for 24 h; then the RNA was extracted. The gene sequence is shown in Table 3.

### 4.11. Statistical Analysis

Statistical analysis was performed using SPSS 23.0 software program (Chicago, IL, USA) and GraphPad Prism 8.2.0 software (GraphPad Software, Santiago, CL, USA). The differences between the two groups or multiple groups were calculated using a t-test and one-way analysis of variance (ANOVA). The difference was considered statistically significant at *p* < 0.05 and highly significant at *p* < 0.01.

## 5. Conclusions

In conclusion, this study demonstrated that DXXK can treat DOX-induced cardiotoxicity through the AMPK-mediated energy protection pathway and that DXXK participates in the process of regulating autophagy in cardiomyocytes through the non-AMPK pathway. DXXK can be used as an energy protector of the heart to fight against DOX-induced cardiotoxicity, which is very necessary to be fully verified in the future.

## Figures and Tables

**Figure 1 ijms-24-00897-f001:**
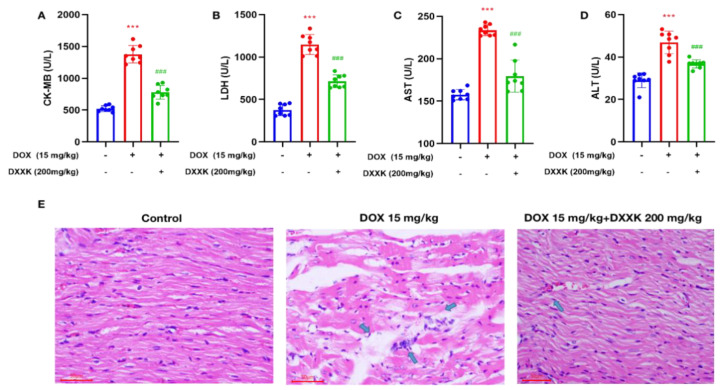
DXXK protects against heart injury caused by DOX. (**A**) CK-MB; (**B**) LDH; (**C**) ALT; (**D**) AST; (**E**) Representative photomicrographs for HE staining of heart histological changes. Cell vacuoles, hydropic degenerations and inflammatory cell infiltration areas are indicated by arrows. *** *p* < 0.001 compared with the control group. ^###^
*p* < 0.001 compared with the DOX group.”+”indicates that drugs are added, and “-” indicates that drugs are not added.

**Figure 2 ijms-24-00897-f002:**
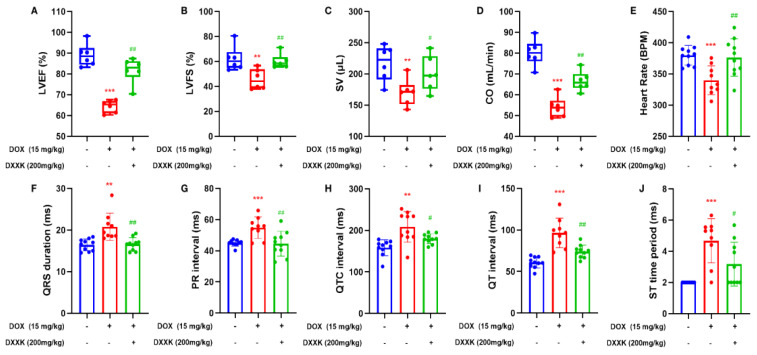
DXXK can improve DOX-induced cardiac function decline and electrical signal conduction block in rats. (**A**) LVEF; (**B**) LVFS; (**C**) CO; (**D**) HR; (**E**) QRS duration; (**F**) PR interval; (**G**) QTC interval; (**H**) QT interval; (I) QT interval; (**J**) ST time period. ** *p* < 0.01 and *** *p* < 0.001 compared with the control group. ^#^
*p* < 0.05 and ^##^
*p* <0.01 compared with the DOX group. ”+”indicates that drugs are added, and “-” indicates that drugs are not added.

**Figure 3 ijms-24-00897-f003:**
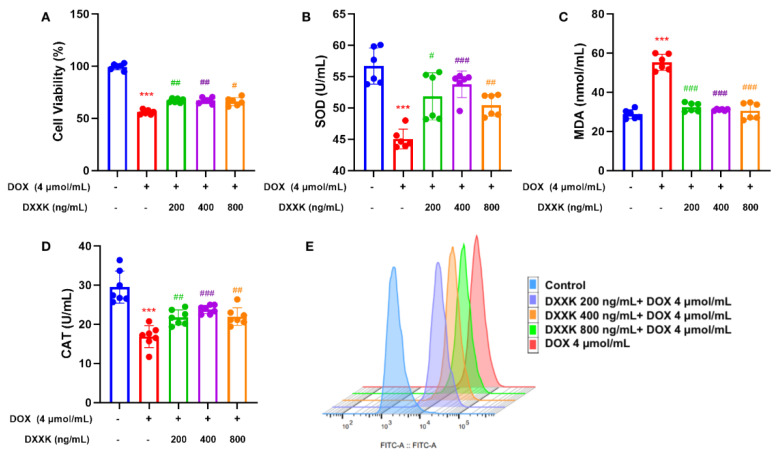
DXXK can protect H9c2 cells from DOX-induced injury. (**A**) Cell viability; (**B**) SOD; (**C**) MDA; (**D**) CAT; (**E**) ROS. *** *p* < 0.001 compared with the control group. ^#^
*p* < 0.05, ^##^
*p* < 0.01 and ^###^
*p* < 0.0001 compared with the DOX group. ”+”indicates that drugs are added, and “-” indicates that drugs are not added.

**Figure 4 ijms-24-00897-f004:**
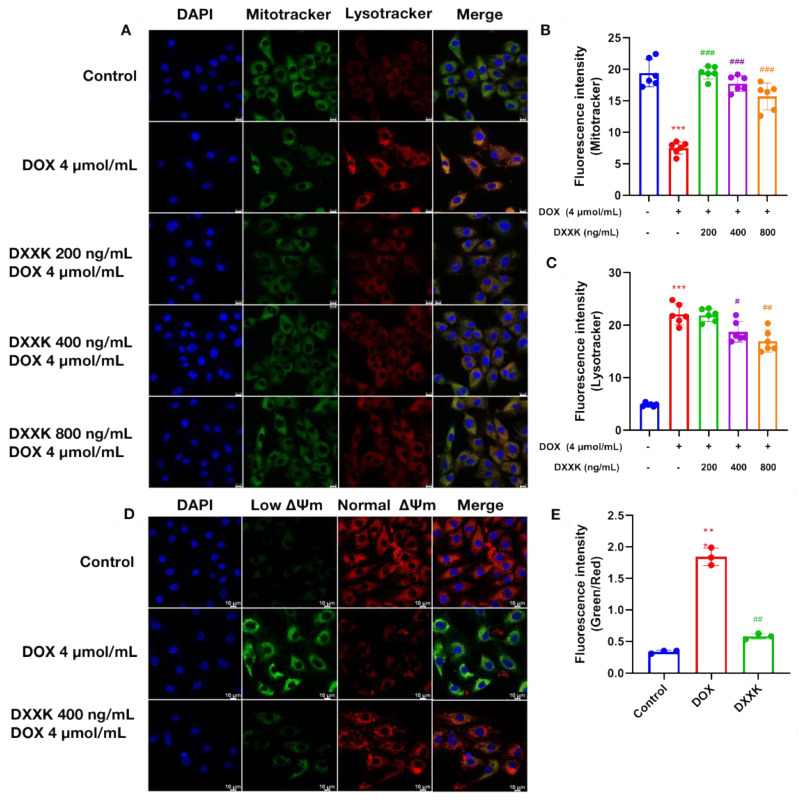
DXXK protects myocardial mitochondria from DOX-induced damage. (**A**) Observation of the number of mitochondria and lysosomes by laser confocal microscopy. Green fluorescence intensity represents mitochondria. Red fluorescence intensity represents the number of autophagosomes. (**B**) Green fluorescence intensity; (**C**) red fluorescence intensity; (**D**) observation of the mitochondrial membrane potential by laser confocal microscopy. (**E**) The higher the ratio of green fluorescence intensity to red fluorescence intensity, the lower the mitochondrial membrane potential. ** *p* < 0.01 and *** *p* < 0.001 compared with the control group. ^#^
*p* < 0.05, ^##^
*p* < 0.01 and ^###^
*p* < 0.001 compared with the DOX group.

**Figure 5 ijms-24-00897-f005:**
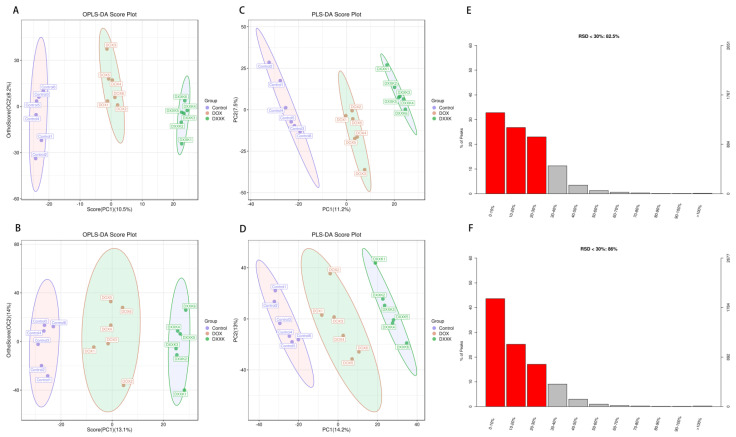
Metabolic profiles and differentiation of the control, DOX and DXXK groups by multivariate analysis. OPLS−DA score scatter plots in the ESI+ mode (**A**) and ESI− mode (**B**); S−plot of the PLS−DA model in the ESI+ mode (**C**) and ESI− mode (**D**); RSD in the ESI+ mode (**E**) and ESI- mode (**F**).

**Figure 6 ijms-24-00897-f006:**
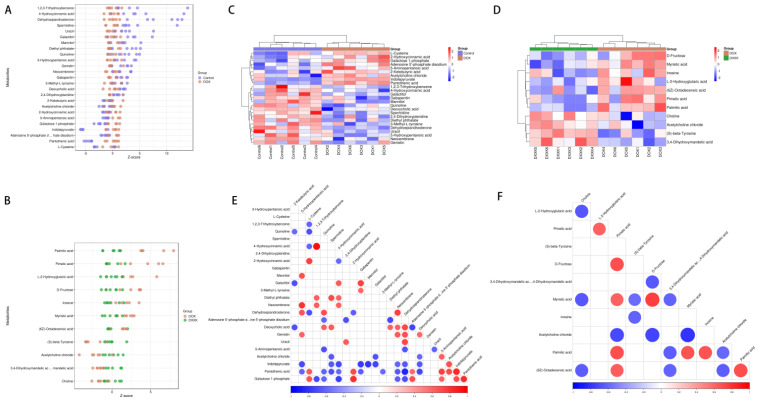
Identification of potential biomarkers of DOX−induced cardiotoxicity and DXXK cardioprotection. (**A**) Changes of differential metabolites between the control group and DOX group. (**B**) Changes of differential metabolites between the DOX group and DXXK group. (**C**) The cluster heatmap of metabolites in the control group and DOX group. (**D**) The cluster heatmap of metabolites both in the DOX group and DXXK group. (**E**) Correlation analysis of different metabolites between the control group and DOX group. (**F**) Correlation analysis of different metabolites between the DOX group and DXXK group.

**Figure 7 ijms-24-00897-f007:**
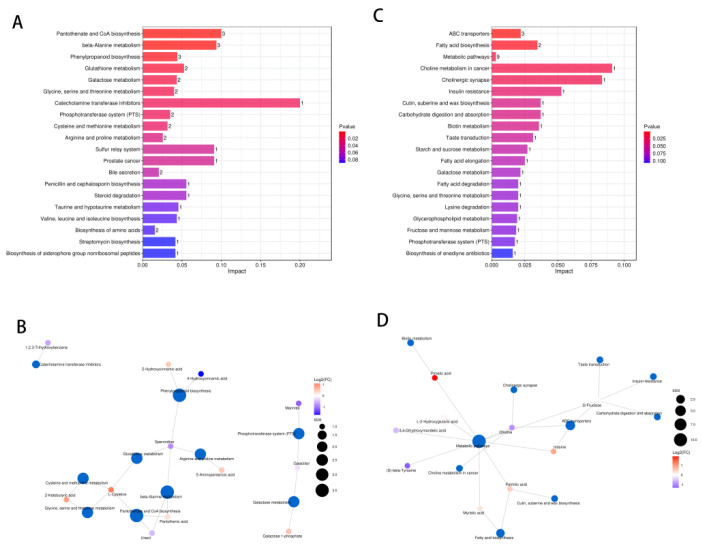
Pathway analysis of DXXK protecting myocardial cells from DOX-induced injury. (**A**) Enrichment analysis of signaling pathways between the control group and DOX group. (**B**) Enrichment analysis of signal pathway between the DOX group and DXXK group. (**C**) The “Pathway-Metabolite” network of DOX-induced myocardial cell damage. (**D**) The “Pathway-Metabolite” network of DXXK protecting myocardial cell damage.

**Figure 8 ijms-24-00897-f008:**
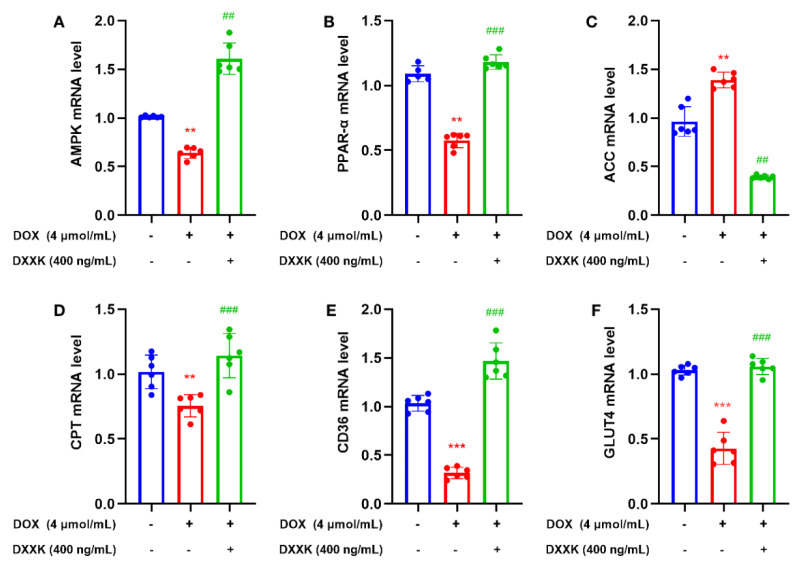
DXXK can promote the expression of key target mRNAs in the myocardial energy metabolism pathway. (**A**) The mRNA level of AMPK. (**B**) The mRNA level of PPAR-α. (**C**) The mRNA level of ACC. (**D**) The mRNA level of CPT. (**E**) The mRNA level of CD36. (**F**) The mRNA level of GLUT4. ** *p* < 0.01 and *** *p* < 0.001 compared with the control group. ^##^
*p* < 0.01 and ^###^
*p* < 0.001 compared with the DOX group. ”+”indicates that drugs are added, and “-” indicates that drugs are not added.

**Figure 9 ijms-24-00897-f009:**
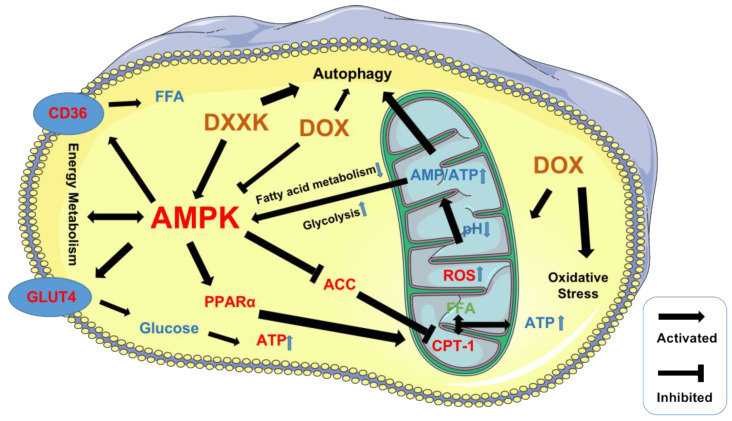
The mechanism of energy damage of cardiac myocytes induced by DOX and energy protection of DXXK.

**Table 1 ijms-24-00897-t001:** Potential differential metabolites of DOX-induced myocardial cell injury.

ID	Name	*m/z*	rt	ppm	Formula	KEGG
M121T90	L-Cysteine	120.9811	89.7	0.740305618	C_3_H_7_NO_2_S	C00097
M218T245	Pantothenic acid	218.1026	245.1	1.300497289	C_9_H_17_NO_5_	C00864
M111T750	Uracil	111.0181	750.2	5.46386555	C_4_H_4_N_2_O_2_	C00106
M146T958_2	Spermidine	145.9846	957.7	14.71119161	C_7_H_19_N_3_	C00315
M183T469	Galactitol	183.0864	469	0.485376675	C_6_H_14_O_6_	C01697
M102T90_2	2-Ketobutyric acid	102.0342	89.7	0.014065686	C_4_H_6_O_3_	C00109
M117T877	5-Aminopentanoic acid	116.9259	876.8	10.96036025	C_5_H_11_NO_2_	C00431
M164T198	2,4-Dihydroxypteridine	163.9405	197.8	2.624555689	C_6_H_4_N_4_O_2_	C03212
M182T979	Mannitol	181.978	978.7	4.796869979	C_6_H_14_O_6_	C00392
M288T647	Dehydroepiandrosterone	288.2897	647.2	0.540973302	C_19_H_28_O_2_	C01227
M432T719	Genistin	432.238	718.6	0.472864253	C_21_H_20_O_10_	C09126
M119T339	5-Hydroxypentanoic acid	119.0689	339	11.55633419	C_5_H_10_O_3_	C02804
M202T784	Indolepyruvate	201.9045	784	4.770939835	C_11_H_9_NO_3_	C00331
M127T114	1,2,3-Trihydroxybenzene	127.0389	113.6	0.598241956	C_6_H_6_O_3_	C01108
M259T91	Galactose 1-phosphate	259.0217	91.3	2.702476279	C_6_H_13_O_9_P	C00103

**Table 2 ijms-24-00897-t002:** Potential metabolites of DXXK in treating DOX-induced myocardial cell injury.

ID	Name	*m/z*	rt	ppm	Formula	KEGG
M181T782	D-Fructose	181.0146	782.3	3.073909779	C_6_H_12_O_6_	C00095
M104T817	Choline	104.1079	817.4	3.842167597	C_5_H_14_NO	C00114
M269T264	Inosine	269.0873	263.6	2.601386242	C_10_H_12_N_4_O_5_	C00294
M160T81	Pimelic acid	159.9675	80.9	10.19875742	C_7_H_12_O_4_	C02656
M229T648_1	Myristic acid	229.1802	647.7	3.967308328	C_14_H_28_O_2_	C06424
M149T215	L-2-Hydroxyglutaric acid	149.0249	214.5	9.097060136	C_5_H_8_O_5_	C03196
M185T82	3,4-Dihydroxymandelic acid	184.9844	82.3	6.298831399	C_8_H_8_O_5_	C05580

**Table 3 ijms-24-00897-t003:** Primer sequence.

Gene	Sequence (5′-3′)	Product Size	ID
AMPK	TACCTCGCCTCCAGTCC GTGCTTTGGGGCTGTCT	121	78,975
PPAR-α	TCATCACCCGAGAGTTCC TCCAGTTCGAGGGCATT	100	25,747
CD36	GAACCAGGCCACATAGAAAG ACCAATAACGGCTCCAGTAA	138	29,184
ACC	TTGGACAACGCCTTCAC GCAGCCCATTACTTCATCA	102	60,581
CPT-1	TGTGGCTTGCTGTATTTGA TGACTGGGTGGGATTAGAA	172	25,757
GLUT4	CAGGATGAAGGAAACAGCA GCAGGACGGGAGAAAAG	150	25,139

## Data Availability

All figures and data used to support this study are included within this article.

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
