# Peer review of "Targeting Energy Protection as a Novel Strategy to Disclose Di’ao Xinxuekang against the Cardiotoxicity Caused by Doxorubicin"

_ijms, 2023, doi:10.3390/ijms24020897_

Round 1

Reviewer 1 Report

This is a thorough study on cardioprotective effect of DXXK. After careful analysis of results obtain, I agree with the authors' conlusions. Methodology, results and interpreation does not raise significant concerns. This article, however, is full of small editorial mistakes, which should be removed before its publication. Moreover, I think it should be corrected by professional English proofreader or native English speaker. I am not qualified in this field, yet even I can see that the language and style require some correction.

I've attached the PDF with minor comments and corrections.

Author Response

Dec. 25, 2022

Dear Expert Reviewer,

Thank you very much for the prompt review process and excellent comments. We greatly appreciate the time and efforts which you have spent on it. We are submitting the revised manuscript entitled “Targeting energy protection as novel strategy to disclose Di'ao Xinxuekang against the cardiotoxicity caused by doxorubicin(ID: ijms-2093672) to International Journal of Molecular Sciences.

We have carefully considered your comments and suggestions, and we have revised the manuscripts based on your comments and carefully checked throughout the manuscript. Firstly, we redraw and replaced some non-standard pictures according to your suggestions (Figure. 4, page 7). Secondly, we have supplemented the necessary information missing in the manuscript that you have proposed (such as the explanation of abbreviations, the manufacturer of materials, etc.) Finally, I will give the following answers to your questions in the comments. Our responses to the comments (in blue) are shown below (in red).

  1. This article, however, is full of small editorial mistakes, which should be removed before its publication.

Response: Thank you for your reminder. We have checked all the contents of the manuscript again and revised it according to your suggestions.

  1. Moreover, I think it should be corrected by professional English proofreader or native English speaker.

Response: We feel great thanks for your professional review work on our article. We have asked professional English proofreader to check the language of the manuscript.

  1. Why you've changed the box plots to bar plots? (Figure. 2, page 4)

Response: Thank you very much for your suggestions. We want to use the box plots and bar plots to distinguish between hemodynamic indicators and cardiac electrical signal transmission indicators (Figure. 2, page 4). Different types of indicators are presented in different pictures, which has also appeared in many articles.

  • Tadokoro T, Ikeda M, Ide T, Deguchi H, Ikeda S, Okabe K, Ishikita A, Matsushima S, Koumura T, Yamada KI, Imai H, Tsutsui H. Mitochondria-dependent ferroptosis plays a pivotal role in doxorubicin cardiotoxicity. JCI Insight. 2020 May 7;5(9):e132747. doi: 10.1172/jci.insight.132747. PMID: 32376803; PMCID: PMC7253028.
  • Wen J, Zhang L, Wang J, Wang J, Wang L, Wang R, Li R, Liu H, Wei S, Li H, Zou W, Zhao Y. Therapeutic effects of higenamine combined with [6]-gingerol on chronic heart failure induced by doxorubicin via ameliorating mitochondrial function. J Cell Mol Med. 2020 Apr;24(7):4036-4050. doi: 10.1111/jcmm.15041. Epub 2020 Feb 19. PMID: 32073745; PMCID: PMC7171398.

Thank you for all the valuable and helpful comments and suggestions.

Best regards,

Jianming Wu

Reviewer 2 Report

Dear Authors,

Cardiotoxicity caused by anthracyclines is a real challenge in oncology in terms of both: early detection and treatment.  

I have red your paper with interest, please find my comments below

Abstract: please add more details in place of very general satements. As that is another publication regarding in vitro studies and animal model with the same compound and the same indication one might think you just publish the same again.

Introduction: while it is easy to believe that anthracyclines are used in majority of patients with hematological malignancies it is very unlikely they are used in majority of patients with solid tumors. Anthracyclines are not used (or very occasionally used) in lung ca, GI malignancies (apart from gastric ca). 

That would be great to see some graph/figure with proposed anthracycline-induced cardiotoxicity mechanisms.

Discussion and introduction: please kindly discuss and cite your previous publication

Li X, Liang J, Qin A, Wang T, Liu S, Li W, Yuan C, Qu L, Zou W. Protective effect of Di'ao Xinxuekang capsule against doxorubicin-induced chronic cardiotoxicity. J Ethnopharmacol. 2022 Apr 6;287:114943. doi: 10.1016/j.jep.2021.114943. Epub 2021 Dec 24. PMID: 34954266.

That would be also good to inform about your future plans regarding that topic.

Author Response

Dec. 25, 2022

Dear Expert Reviewer,

Thank you very much for the prompt review process and excellent comments. We greatly appreciate the time and efforts which you have spent on it. We are submitting the revised manuscript entitled “Targeting energy protection as novel strategy to disclose Di'ao Xinxuekang against the cardiotoxicity caused by doxorubicin(ID: ijms-2093672) to International Journal of Molecular Sciences.

We have carefully considered your comments and suggestions, and we have revised the manuscripts based on your comments and carefully checked throughout the manuscript. Our point-by-point responses to the comments (in blue) are shown below (in red).

  1. Abstract: please add more details in place of very general satements. As that is another publication regarding in vitro studies and animal model with the same compound and the same indication one might think you just publish the same again.

Response: Thank you very much for your suggestions. We have added some content to the abstract to highlight the uniqueness of this study (line 18-19, page1). In addition, the purpose of this study was to explore the mechanism of adriamycin induced energy depletion in rat cardiomyocytes and the effect of Di'ao Xinxuekang on its improvement. This is a new study, which is different from the previous study on Di'ao Xinxuekang's effect on doxorubicin induced ferroptosis in mouse cardiomyocytes.

  1. Introduction: while it is easy to believe that anthracyclines are used in majority of patients with hematological malignancies it is very unlikely they are used in majority of patients with solid tumors. Anthracyclines are not used (or very occasionally used) in lung ca, GI malignancies (apart from gastric ca).

Response: Thank you very much for your suggestions, Indeed, as you said, doxorubicin are not used (or very occasionally used) in lung ca, GI malignancies (apart from gastric ca). We have revised the expression of relevant contents. But doxorubicin can be used in soft tissue sarcoma, testicular cancer and ovarian cancer.

[1] Scheulen ME, Seeber S, Schilcher RB. Sequential combination chemotherapy with vinblastine-bleomycin and doxorubicin-cis-dichlorodiammineplatinum(II) in disseminated nonseminomatous testicular cancer. Cancer Treat Rep. 1980 Apr-May;64(4-5):599-609. PMID: 6159077.

[2] Hamanishi J, Takeshima N, Katsumata N, Ushijima K, Kimura T, Takeuchi S, Matsumoto K, Ito K, Mandai M, Nakai H, Sakuragi N, Watari H, Takahashi N, Kato H, Hasegawa K, Yonemori K, Mizuno M, Takehara K, Niikura H, Sawasaki T, Nakao S, Saito T, Enomoto T, Nagase S, Suzuki N, Matsumoto T, Kondo E, Sonoda K, Aihara S, Aoki Y, Okamoto A, Takano H, Kobayashi H, Kato H, Terai Y, Takazawa A, Takahashi Y, Namba Y, Aoki D, Fujiwara K, Sugiyama T, Konishi I. Nivolumab Versus Gemcitabine or Pegylated Liposomal Doxorubicin for Patients With Platinum-Resistant Ovarian Cancer: Open-Label, Randomized Trial in Japan (NINJA). J Clin Oncol. 2021 Nov 20;39(33):3671-3681. doi: 10.1200/JCO.21.00334. Epub 2021 Sep 2. PMID: 34473544; PMCID: PMC8601279.

  1. That would be great to see some graph/figure with proposed anthracycline-induced cardiotoxicity mechanisms.

Response: We feel great thanks for your professional review work on our article. We briefly describe the mechanism of cardiotoxicity caused by doxorubicin (As a representative of anthracycline drugs) in introduction (This part is marked with a yellow mark, line 64-68, page 2). In addition, we attach two articles detailing the pathological mechanism of anthracycline induced cardiotoxicity for your reference. At the same time, we also attach the key mechanism diagram in the literature for your convenience.

[1] Cardinale D, Iacopo F, Cipolla CM. Cardiotoxicity of Anthracyclines. Front Cardiovasc Med. 2020 Mar 18;7:26. doi: 10.3389/fcvm.2020.00026. PMID: 32258060; PMCID: PMC7093379.

[2] Christidi E, Brunham LR. Regulated cell death pathways in doxorubicin-induced cardiotoxicity. Cell Death Dis. 2021 Apr 1;12(4):339. doi: 10.1038/s41419-021-03614-x. PMID: 33795647; PMCID: PMC8017015.

Fig. 1 Doxorubicin-induced ferroptosis (Can be found in the attachment WORD)

Fig. 2 Doxorubicin-induced apoptosis (Can be found in the attachment WORD)

  1. Discussion and introduction: please kindly discuss and cite your previous publication

Response: Thank you for your reminder. We have added relevant content according to your suggestions (line 383-387, page 16). Previous studies have found that Di'ao Xinxuekang (DXXK) can improve doxorubicin induced cardiotoxicity in mice by inhibiting ferroptosis. In this study, a model of doxorubicin induced cardiotoxicity was established in rats, and a new interpretation of DXXK's cardioprotective effect was made from the perspective of energy metabolism.

Thank you for all the valuable and helpful comments and suggestions.

Best regards,

Jianming Wu
